# Microbial-Derived Tryptophan Catabolites, Kidney Disease and Gut Inflammation

**DOI:** 10.3390/toxins14090645

**Published:** 2022-09-18

**Authors:** Avra Melina Madella, Jeroen Van Bergenhenegouwen, Johan Garssen, Rosalinde Masereeuw, Saskia Adriana Overbeek

**Affiliations:** 1Department of Pharmaceutical Sciences, Utrecht University, Universiteitsweg 99, 3584 CG Utrecht, The Netherlands; 2Danone Nutricia Research, Uppsalalaan 12, Utrecht Science Park, 3584 CT Utrecht, The Netherlands

**Keywords:** CKD, intestinal inflammation, tryptophan derivatives, indoles, indoxyl sulfate, biotics, SCFAs

## Abstract

Uremic metabolites, molecules either produced by the host or from the microbiota population existing in the gastrointestinal tract that gets excreted by the kidneys into urine, have significant effects on both health and disease. Tryptophan-derived catabolites are an important group of bacteria-produced metabolites with an extensive contribution to intestinal health and, eventually, chronic kidney disease (CKD) progression. The end-metabolite, indoxyl sulfate, is a key contributor to the exacerbation of CKD via the induction of an inflammatory state and oxidative stress affecting various organ systems. Contrastingly, other tryptophan catabolites positively contribute to maintaining intestinal homeostasis and preventing intestinal inflammation—activities signaled through nuclear receptors in particular—the aryl hydrocarbon receptor (AhR) and the pregnane X receptor (PXR). This review discusses the origins of these catabolites, their effect on organ systems, and how these can be manipulated therapeutically in the future as a strategy to treat CKD progression and gut inflammation management. Furthermore, the use of biotics (prebiotics, probiotics, synbiotics) as a means to increase the presence of beneficial short-chain fatty acids (SCFAs) to achieve intestinal homeostasis is discussed.

## 1. Introduction

Uremic syndrome is defined as the terminal clinical manifestation of chronic kidney disease (CKD), largely caused by increased retention of a variety of uremic metabolites [1]. Uremic metabolites are molecules resulting from either host or bacterial metabolic processes and are normally removed from the organism by renal excretion pathways through urine. When kidney function declines, the uremic metabolites’ increased systemic concentrations lead to various deleterious effects, including the progression of CKD as well as cardiovascular disease; for this reason, they are collectively termed uremic toxins [2,3].

The European Uremic Toxin workgroup (https://www.uremic-toxins.org/ (accessed on 1 August 2022) has classified these molecules according to their physicochemical properties in relation to their removal patterns in three categories; *(a)* small water-soluble molecules (MW ≤ 500 Da) that can be removed by conventional dialysis such as creatinine and urea, *(b)* middle weight molecules (MW ≥ 500 Da) such as β2-microglobulin that can be more effectively removed by peritoneal dialysis due to the larger pore size and longer dialysis duration, and *(c)* protein-bound compounds such as indoxyl sulfate (IS) and p-cresol sulfate (PCS) that are firmly bound to blood plasma proteins (predominantly albumin) and are difficult to remove by conventional haemodialysis [1,4].

Another method of classification is according to their origin; (a) endogenous metabolism, (b) microbial metabolism, and (c) exogenous intake. The classification of these metabolites based on origin is helpful for the identification of therapeutic targets for CKD.

The majority of these metabolites are by-products from endogenous metabolic processes, although it is becoming more apparent that metabolites from microbial metabolism are also of crucial importance [5]. Metabolites produced by microbial metabolism are further classified into three categories (Figure 1), depending on the involvement of microbial species in the synthesis of these specific metabolites. Specifically, they are classified as *(a)* metabolites that are produced by bacteria from dietary sources, *(b)* metabolites that are produced by the host and biochemically modified by gut bacteria, and *(c)* metabolites that are synthesised de novo by gut microbes [6]. Metabolites that are produced by bacteria from dietary components include short chain fatty acids (SCFAs), polyamines, and indole derivatives, while metabolites that are produced by the host and biochemically modified by gut bacteria include secondary bile acid and taurine [7,8,9]. Finally, metabolites that are synthesised de novo by gut microbes include polysaccharides A (PSA) and adenosine triphosphate (ATP) [10,11].

An increasing number of studies demonstrate the relevance of the gut microbiome to health and disease, with a special role for microbial metabolites [12]. Bacterial metabolism has great significance in the breakdown of dietary products to metabolites with both beneficial effects as well as potentially harmful solutes associated with the development and progression of the disease [13]. Metabolites produced by microbiota form an essential component of the crosstalk that occurs between the gut and the kidneys. Interactions between the two organs are bidirectional and are collectively termed as the gut-kidney axis [14]. Alterations in homeostasis lead to the induction of dysbiosis, which can play a role in CKD and related cardiovascular diseases.

Uremic metabolites, predominantly originating from proteolytic fermentation, have now been associated with the development of CKD. Increased accumulations of these metabolites, such as IS, show a strong correlation with more severe clinical outcomes and higher mortality risk [15,16,17]. Furthermore, IS was found to induce inflammation by stimulating the activation of the nuclear factor kappa B (NF-κB) pathway and plasminogen activator inhibitor type 1, resulting in nephrotoxic effects in renal proximal tubular cells [18]. On the other hand, kidney failure leads to an accumulation of uremic toxins systemically. The increased presence of urea in the intestine then further exacerbates gut dysbiosis, leading to the synthesis of more IS, which then feeds back into the progressively worsening renal state. Elevated urea levels impact gut permeability and inflammation by two pathways; *(a)* causing a breakdown of the tight junction protein barrier in the epithelial barrier, making the barrier become leakier and contributing to endotoxemia, the low-grade level inflammation that exists in CKD patients; and *(b)* modifying the microbial flora [19,20]. Once urea has diffused into the gut lumen, it is metabolised by gut bacterial urease to ammonia, which then alters the pH in the gut lumen. The altered pH then acts as a selection pressure, causing an expansion in bacterial families with the capability to break down urea and a decreased presence of other bacterial families [21].

However, not all metabolites synthesised by bacterial metabolism are associated with negative health outcomes and CKD progression. In contrast to the evidence shown for uremic metabolites, bacteria-produced SCFAs have been associated with CKD-protective effects. SCFAs have been shown to beneficially modulate fibrotic and epigenetic factors in research models of CKD, as well as to reduce blood pressure, inflammation, and immunity [22].

The aim of this review is to examine uremic metabolites that have been synthesised by bacteria from dietary sources. Given the involvement of certain tryptophan catabolites, such as IS on CKD, the focus will be given to indole and its derivatives, including special consideration on their function on gut health and inflammation. Although the investigation of the relevance of the gut microbiome manipulation with respect to CKD is still in the early explorative stages, the potential benefits of microbiome management, e.g., the use of prebiotics, probiotics, and synbiotics as therapeutic interventions will also be explored.

## 2. Origins and Distribution of Tryptophan Catabolites

Tryptophan is an essential amino acid naturally provided by diet and an important precursor for many different bioactive metabolites [23]. As bacteria depend on tryptophan catabolites such as indole and indole-based compounds for their intercellular signalling, both Gram-positive as well as Gram-negative bacteria express an abundance of enzymes that are capable of processing tryptophan into molecules that would support indole signalling [24,25]. Different microbial species possess different catalytic enzymes, necessitating cooperation between bacterial species for the synthesis of particular metabolites. Adding further complexity to the matter, the characterisation and identification of bacterial species possessing certain catalytic enzymes remain to be elucidated in many cases [26]. In general, bacteria metabolise tryptophan using the indole pyruvate pathway utilising three different key enzymes to produce a variety of metabolites [27,28]. The enzyme tryptophan dehydrogenase (TrpD) converts tryptophan into tryptamine which is a precursor for the metabolites indole-3-ethanol (IE), indole-acetate, and indole-3-acetic acid (IAA), which can be further metabolised into 3-methylindole (Skatole) and indole-3-aldehyde (IAld). Bacterial-derived tryptophanase converts tryptophan into IAld, IAA, indole-3-propionic acid (IPA), and indole, which is processed by the liver into indoxyl and then sulfated into IS and indole-3-carboxylic acid. The aromatic amino acid aminotransferase-dependent pathway converts tryptophan into indole-3-pyruvate (IPγA), which can be further processed into indole-3-lactic acid (ILA), indoleacrylic acid (IAcr), and IAA (Figure 2).

These indole products, with the exception of IS, are of crucial importance in the potential role of the microbial community for drug resistance, biofilm formation, toxicity, plasmid stability, and spore formation [29,30,31,32]. Additionally, indoles also suppress inflammation, regulate gut insulin secretion, and strengthen the efficacy of intestinal epithelial cells [33].

## 3. Tryptophan Catabolites, AhR, and PXR

Over the past years, it has become clear that the aryl hydrocarbon receptor (AhR) and the pregnane X receptor (PXR) are the primary targets for interaction with bacterial-produced tryptophan catabolites [35,36]. AhR is a ligand-activated transcription factor whose activity is tightly controlled through repressors that prevent AhR-transcriptional activity and AhR-activated negative feedback loops. AhR is expressed by a variety of different cell types, among which are epithelial cells and immune cells, and its expression is increased in inflammatory and malignant disorders. Activation of AhR is an important factor in the maintenance of tissue homeostasis and immunity and functions as a key sensor allowing immune cells to adapt to environmental conditions, such as air pollution [37]. Decreased activation, as well as hyperactivation of AhR, may contribute to the pathogenesis of multiple human diseases such as inflammatory bowel disease, autoimmune disorders, and cancer [38,39,40]. In the context of CKD, elevated levels of uremic tryptophan catabolites, especially IS, may lead to dysregulation of AhR-activity, promoting vascular disease, renal tissue fibrosis, and pro-inflammatory responses [41].

Similar to AhR, PXR is a nuclear receptor with a high substrate diversity of ligands and its activation leads to the expression of drug-metabolising enzymes and drug efflux transporters. For this reason, PXR activation plays an important role in detoxification processes protecting the body from harmful foreign toxicants or endogenous toxic substances [42]. PXR is mainly expressed by intestinal epithelial and liver cells but also by immune cells. In the intestine, PXR is important as a central regulator of epithelial barrier function; the absence of PXR or reduces PXR expression increases intestinal susceptibility toward xenobiotic injury and increases intestinal inflammatory responses [43]. Although PXR is expressed in B-cells, T-cells, dendritic cells, and monocytes [44,45] the exact role of PXR in these cells remains to be defined. However, it could include anti-inflammatory functions as PXR exhibits significant repressive interactions with nuclear factor kappa B (NFκB), which is a master transcription factor of inflammation [46,47].

The interaction between PXR and NFκB seems of special importance in managing intestinal inflammatory responses [48,49]. In the context of CKD, PXR-activation confers protective functions against kidney damage, while a diminished PXR activation might reduce the expression of drug-metabolising enzymes, changing the pharmacokinetics in CKD [50,51]. 

Analysis of faecal content of tryptophan catabolites indicates that the presence of specific bacterial species is necessary for the occurrence of these catabolites, and their levels depend on diet and/or microbiota composition [38,52]. Moreover, physiologically relevant concentrations of these catabolites in either serum or feces are sufficient to induce AhR activation to different extents, as also shown in Table 1 [52]. 

However, there exists a significant difference between affinity and activation between mouse and human AhR, which could result in a 10-fold difference in ligand-binding affinity between the mouse AhR and human AhR for a diverse range of AhR ligands [53]. Tryptophan catabolite interactions with AhR and PXR were investigated in detail by the Dvorak group [35,36,54]. While all catabolites tested were capable of dose-dependently binding to and activating AhR and PXR in luminescence-based reporter assays, not all of them were able to induce the expression of *CYP1A1* mRNA, a sensitive marker for AhR-activation, or *CYP3A4* mRNA, a sensitive marker for PXR-activation. This discrepancy could potentially be explained by the interactions between AhR and PXR. Activation of PXR has been shown to prevent the binding of AhR to its target genes and subsequently suppress gene expression [55]. On the other hand, some synergy between AhR and PXR could potentially be expected as indole co-incubations significantly increased IPA and IAA-induced PXR activation, although the involvement of AhR was not confirmed [43]. Interestingly, specific catabolites also showed strong antagonistic properties against one or more of the prototypical AhR-ligands 2,3,7,8-tetrachlorodibenzo-p-dioxin (TCDD), benzo(a)pyrene (BaP) and 6-formylindolo [3,2-b]carbazole (FICZ). Antagonism was observed at the level of AhR activation but also at the level of inhibition of *CYP1A1* expression. The catabolite indole proved to be the most proficient antagonist of AhR-activation, confirming previously obtained results using a different methodology [36,56]. Antagonistic activity has been shown by other tryptophan catabolites, but results are rather ambiguous at this stage.

While more and more data become available on the cellular effects of isolated catabolites, it remains difficult to predict the result of catabolite mixtures as present in faecal material or serum/systemically. This is mainly due to the ongoing discovery of novel catabolites and their receptors but also due to the promiscuity of the nuclear receptors and their complex interactions among themselves and with NF-κΒ and the inflammasome [57,58,59,60].

## 4. The Role of Tryptophan Catabolites in Gut Health and CKD

Tryptophan catabolites significantly influence intestinal function by affecting its barrier efficacy through modulating immune responses, permeability and tight junction formation, and mucus synthesis [33]. Although tryptophan derivatives are essential in improving and maintaining intestinal health, they can still be responsible for deleterious effects throughout organ systems. The most widely studied example is IS, a pro-inflammatory metabolite that affects various organs and systems and contributes to CKD-associated comorbidities [61].

Over the years, it has become evident that patients with CKD suffer from dysbiotic microbiota [62]. CKD-induced changes in the microbiota not only impacts microbiota composition but also the array of microbiota-produced metabolites [63]. To maintain proper intestinal functioning and homeostasis, the host depends on carefully controlled interactions between microbes, microbial metabolites, epithelial functioning, and mucosal immunity. In patients with CKD, this precarious balance is disrupted, leading to reduced intestinal barrier function and increased intestinal inflammation [64,65]. It can be hypothesised that beneficial tryptophan catabolites may improve CKD through their contribution to intestinal homeostasis by skewing the balance towards a more eubiotic microbiota composition. In addition, reduction in deleterious tryptophan derivatives will also contribute to improvement in disease. The effects of these metabolites on gut health are discussed in more detail below and in Table 2.


Tryptamine:


Tryptamine, is a neurotransmitter acting on the enteric nervous system and modulating intestinal homeostasis whilst also affecting intestinal motility [66]. Additionally, it can prevent the invasion and colonisation ability of pathogenic species such as *Salmonella enterica serovar Typhimurium* [67]. It has also shown inhibitory activity against the enzyme Indoleamine 2,3-dioxygenase 1 (IDO1), which has implications for immune escaping and immunomodulation [68]. 


IAA


The effects of IAA have been characterised as ambivalent with regard to CKD. IAA has been found to act both through PXR and AhR. IAA has been found to activate AhR on dendritic cells, leading to the production of the anti-inflammatory cytokine IL-22 [69]. Through its affinity for PXR, IAA was found to promote the synthesis and accumulation of IL-35^+^ B regulatory (B_reg_) cells in the intestine, which are key regulators for a myriad of diseases whilst IL-35 expression is dysregulated in inflammatory autoimmune diseases, most notably IBD [70]. Even though its effects on the maintenance of intestinal homeostasis indicate a beneficial effect, IAA has also been associated with the exacerbation of fibrosis, cardiovascular disease, thrombogenicity, and inflammation in CKD [61]. Further analysis is required to disseminate the mechanisms by which IAA might be beneficial for the maintenance of intestinal homeosis but exert deleterious effects in the exacerbation of CKD.


Skatole:


Skatole, also known as 3-methylindole, has a bacteriostatic effect against Gram-negative enterobacteria, while it can also influence the growth and reproduction of other bacterial species by enhancing the growth of bacteria-producing skatole [71]. Its exact effect on intestinal microbes and, consequently, intestinal homeostasis is not fully elucidated yet.


IAld:


Through its activation of AhR, IAld enhances IL-22 transcription and decreases intestinal inflammation whilst contributing to the maintenance of intestinal homeostasis and enhancing intestinal epithelial cell regeneration. Moreover, it has also shown antifungal resistance in both mice and humans while providing colonisation resistance [72].


IPA:


IPA has been found to have an affinity for both AhR and PXR, similar to IAA. By activating AhR, IPA exposure significantly increased the expression of the IL-10 receptor-ligand binding subunit (IL-10R1) on intestinal epithelial cells [73]. IPA also demonstrated an anti-inflammatory effect through its activation of PXR. Activation of PXR by IPA was found to downregulate the expression of TNF-α intestinally whilst also increasing the gene expression for tight junction proteins and causing a fortification of the epithelial barrier [43].


Indole:


Indole has been found to strongly influence various aspects of intestinal homeostasis, with the most notable being bacterial physiology in a concentration-dependent manner. Bacterial movement, persister cell formation, biofilm creation, plasmid stability and cell division have all been factors to be affected by indole [74,75,76,77,78]. Additionally, indole has also been found to decrease the synthesis of pro-inflammatory cytokine CXCL-8 and the expression of NF-κB when that had been activated by TNF-α. Moreover, the secretion of anti-inflammatory cytokine IL-10 was found to occur as a result of indole exposure in the same experimental investigation [79].


IPγA:


IPγA showed an anti-inflammatory effect through AhR activation, indicating a significant anti-inflammatory effect in the murine IBD model, through the modulation of Th1 cell differentiation [80].


ILA:


ILA, through its activation of AhR cells in CD4+ T cells, downregulates the transcription factor ThpoK causing CD4+ T cells to differentiate into CD4+ CD8aa+ double-positive intraepithelial T lymphocytes that possess immunomodulatory functions [81].


IAcr


Finally, IAcr, demonstrated an anti-inflammatory effect, by increasing IL-10 production and decreasing the production of TNF-α in the intestinal mucosa. Roles in the enhancement of mucin production were also found, suggesting a role for indole acrylate in the fortification of the epithelial barrier [82].

According to our knowledge, the tryptophan catabolites indole-3-acetamide and indole-3-ethanol, indole acetate, and indole-3-carboxylic acid do not have established roles in CKD or intestinal inflammation.

**Table 2 toxins-14-00645-t002:** Summary of the various tryptophan catabolites and their effects on both CKD and gastrointestinal homeostasis.

Catabolite	Effect on CKD	Effect on Gastrointestinal Homeostasis
Tryptamine	Potentially beneficial through contribution to intestinal homeostasis	Reduced invasion and colonisation by pathogenic species [83]
IAA	Ambivalent; implicated in both exacerbation mechanisms as well as tissue repair mechanisms	Stimulation of IL-22/STAT3 signalling pathway [70] Induce synthesis of IL35+ B cell production and promotion of anti-inflammatory IL-35 release [84]
	Linked to inflammation, fibrosis, metabolic disorders, cardiovascular disease, thrombogenicity [61] Implicated in tissue repair and cell proliferation as well as anti-inflammatory and anti-oxidant action [85,86]	
Skatole	Potentially beneficial through contribution to intestinal homeostasis	Bacteriostatic for certain species, although not fully characterized [71]
IAld	Potentially beneficial through contribution to intestinal homeostasis Protection against metabolic syndrome [87]	Activation of IL-10R1 receptor in an AhR-dependent manner [72,88]
IPA	Potentially beneficial through contribution to intestinal homeostasis Serum IPA levels depleted with CKD progression [89] Involved in the regulation of endothelial function [90]	Enhancement of the integrity of the intestinal epithelial barrier [91,92] Mucosa maintenance through IL-10R1 receptor in an AhR-dependent manner [72,88] Antimicrobial properties [93] Downregulation of pro-inflammatory TNF-α through TLR4 [88]
Indole	Potentially beneficial through contribution to intestinal homeostasis	Enhancement of the integrity of the intestinal epithelial barrier Enhancement of repair mechanisms involved with epithelial barrier Promotion of goblet cell differentiation all through IL-10 release Anti-inflammatory intestinally [79,94,95,96]
IPγA	Potentially beneficial through contribution to intestinal homeostasis Serum IPA levels depleted with CKD progression [89] Involved in the regulation of endothelial function [90]	Enhancement of the integrity of the intestinal epithelial barrier [27,91] Contribution in gut mucosa immune homeostasis maintenance through IL-10R1 receptor in an AhR-dependent manner [72,88] Antimicrobial properties [93] Downregulation of pro-inflammatory TNF-α through TLR4 [88]
ILA	Potentially beneficial through contribution to intestinal homeostasis	Downregulates the transcription factor ThPOK for CD4+ cells, causing them to differentiate into DPIELs—important for immunomodulation [81]
IAcr	Potentially beneficial through contribution to intestinal homeostasis	Increase in IL-10 production and decrease in TNF-α production Enhancement of mucin production [82]
IS	Roles in cardiovascular disease, inflammation, kidney and heart fibrosis, neurotoxicity, disturbed drug removal, and chronic kidney disease-mineral and bone disorder (CKD-MBD) [61,97,98]	Deleterious for intestinal homeostasis, promotes increased intestinal epithelial barrier disruption, contributing to endotoxemia [22]

## 5. Dietary Interventions Targeting Tryptophan Catabolites

As stated previously, tryptophan catabolites produced by gut microbiota play a significant role in both health and disease, with the prominent example of IS, in CKD and associated comorbidities. Consequently, therapeutic approaches targeting these catabolites should aim to decrease the levels of the deleterious IS catabolites and increase levels of beneficial catabolites that have demonstrated their positive effects in the maintenance of intestinal homeostasis and prevention of inflammation. Strategies to decrease IS involve either removing IS from the circulation or decreasing IS synthesis by reducing protein intake [99]. Reducing IS from the circulation of CKD patients either through haemodialysis or peritoneal dialysis is challenging, as IS is tightly bound to albumin, hindering its removal [100]. Diet interventions with restricted protein consumption have shown a beneficial effect, reducing in IS, and improving bacterial diversity in faecal samples, but significant improvements in kidney function were not observed [101,102,103]. Moreover, the use of adsorbent AST-120, whose function was to trap indole and reduce the synthesis of IS, has shown very mixed results on clinical biomarkers such as slower disease progression and mortality [104].

On the other hand, interventions to increase the presence of beneficial tryptophan catabolites have proven to be quite challenging for various reasons. Primarily, the circulating levels of these tryptophan catabolites have not been published in the literature yet, either for a normal healthy population or patients with CDK. These pathways are extremely complex, with extensive crosstalk and cooperation occurring between various bacterial species. It would therefore be a challenging task to make predictions about the effects of any alterations in these synthetic pathways. Nonetheless, these beneficial catabolites do represent an important therapeutic target for ameliorating intestinal homeostasis and could potentially prevent the exacerbation of CKD. They have been assessed in other conditions, most notably lung conditions, with further research warranted [105,106]. Indole, in particular, has shown a range of beneficial effects for intestinal homeostasis and the ability to activate AhR by itself and be an AhR antagonist in the presence of other AhR agonists. Dysregulated AhR signaling is heavily implicated in the exacerbation, if not the driving seat in CKD progression and its comorbidities, and consequently, a potent antagonist, such as indole, would potentially be able to mitigate the effects of the dysregulated signaling [36,56]. Evidence supporting the use of AhR antagonists for CKD comes from both in vitro and in vivo experiments [107,108]. Inhibition of AhR, either pharmacologically or through murine knock-out models, resulted in improvements in proteinuria, hypoalbuminuria, and hyperlipidemia in nephritic rats, as well as ameliorations in several aspects involved with diabetic nephropathy in murine models [108,109]. It should also be noted that indole might be beneficial in antagonising IS for AhR binding but has also shown significantly toxic effects, and hence, further research is warranted before this can be investigated.

Regardless, CKD does represent a vastly unmet medical need and even though the aforementioned strategies do demonstrate significant promise therapeutically, alternatives should also be considered. An example of such a strategy is biotic intervention by the supplementation of probiotics, prebiotics, and synbiotics, which have been shown in both clinical trials and preclinical models to mitigate intestinal inflammation (shown in Table 3). The term prebiotics refers to the supplementation with nutrients that are selectively used by gut microbiota in order to confer beneficial effects to the host, with examples including complex carbohydrates, oligosaccharides, polyphenols, fructans, and galactans [110]. Probiotics involve the administration of living organisms that also elicit a beneficial effect to the host when administered in adequate concentrations through a variety of mechanisms; catabolism of waste products, immunomodulation, and anti-inflammatory effects, as well as the synthesis of bacteriocins in order to suppress the growth of pathogenic bacteria [111]. Most notable examples of probiotics include *Lactobacillus* or *Bifidobacterium* strains. Synbiotics refer to the combination of pro- and prebiotics that can elicit a positive effect on host health [112]. Given the large number of prebiotics and probiotics available, there are multiple possible combinations that can be studied, with some of the most commonly used combinations including *Lactobacillus GG* and inulin, *Bifidobacteria* and *Lactobacilli* with fructooligosaccharides (FOS) or inulin and *Bifidobacteria* and FOS [113]. Postbiotics also represent a biotic intervention option and have recently been defined by the international scientific association of prebiotics and probiotics as a preparation of inanimate microorganisms and/or their components that confers a health benefit on the host [114]. Assessments on the efficacy of postbiotics on intestinal inflammation or the management of CKD have been very limited, and hence postbiotics will not be discussed in this review.

Biotic interventions may result in the improvement in the dysbiotic gut by increasing the presence of bacterial species capable of producing SCFAs, catabolic products resulting from the saccharolytic fermentation of non-digestible carbohydrates, such as fibre [115]. Bacterial families producing SCFAs include the *Bacteroides* and the *Firnicutes,* although a comprehensive list of all bacterial species with capabilities to produce SCFAs remains to be elucidated [116,117]. As both intracellular and extracellular signalling molecules, SCFAs, can affect intestinal immunity through a variety of mechanisms, and their presence in the intestine is of crucial importance for the maintenance of intestinal homeostasis. Their most important functions include enhancing the integrity of the epithelial barrier function through upregulation of the tight junction proteins, the increased synthesis of antimicrobial peptides, preventing entry of pathogenic bacteria, as well as increasing the production of mucins, the first line of defence in the gastrointestinal mucosa [118,119,120,121,122,123,124,125,126,127].

Given the beneficial roles of SCFAs in the fortification of the epithelial barrier and the prevention of leakage of intestinal contents into the circulation, and thus prevention of endotoxemia, shifting the microbiota composition from a proteolytic fermentation to a saccharolytic fermentation represents an attractive therapeutic strategy to alleviate some of the effects of CKD. Skewing fermentation from a proteolytic nature to a saccharolytic nature would not only increase levels of SCFAs but should also decrease the levels of IS.

A meta-analysis of the effect of dietary fibre in CKD by Chiavaroli et al. revealed a beneficial effect for the supplementation in CKD patients, with the primary endpoints being reductions in serum urea and creatinine, suggesting an improvement in kidney function [128]. Biotic supplementation has also shown a beneficial effect in murine models of CKD. Probiotics [129,130,131], prebiotics [132,133,134], synbiotics [135,136,137] have been assessed in various murine CKD models with promising results and an improvement in various parameters (e.g., inflammatory indexes, bacterial diversity, urea and creatinine levels) has been observed as a result of biotic intervention. Of critical note, kidney function was not investigated in these animal models.

A meta-analysis of randomised clinical trials (RCTs) by McFarlane et al. revealed that there is little evidence to support the use of biotics as an intervention therapy for CKD [138]. An important highlight was also a lack of studies that were lengthy in duration, with intervention times ranging from one week to a maximum of 6 months. A likely explanation for the lack of beneficial effect in CKD, which has also been highlighted by other meta-analyses [139,140], would be that 6 months might not be an adequate timeframe to achieve measurable improvements in kidney function.

**Table 3 toxins-14-00645-t003:** Summary of the effects of various murine models and in vivo studies utilising probiotics, prebiotics, and synbiotics on gut inflammation.

Probiotic Intervention	Effect on Inflammation	Type of Study/Species
Lactobacilli	Inhibition of IL-6 production [141]	Ex vivo in LPS-stimulated mononuclear cells from mice
	Downregulation of NF-κB [142]	SAMP1/Yit mice
	Upregulation of MUC3 and MUC3 mRNA expression [143]	HT29 cells
	Improvement in intestinal barrier integrity by inhibition of epithelial cell apoptosis [144]	Healthy Humans
	Lack of remission maintenance of ulcerative colitis (UC) or Crohn’s disease (CD) [145,146]	UC/CD patients
Bifidobacterium	Suppression of Bacteroides vulgatus growth [147]	Mice
	Attenuation of inflammation in IL-10 knock-out mice [148]	IL-10 knock mice
	Improvement in inflammation in colitis [149]	DSS-induced colitis in mice
	Reduction in histological injury score, ileal tissue weight, myeloperoxidase activity, tissue contents of immunoglobulin, TNF-α, and increased IL-10 secretion [150]	SAMP1/Yit mice
VSL #3 (four strains of Lactobacillus, three strains of Bifidobacterium and one strain of Streptococcus)	Reduction in TNF-α and IFN-γ secretion Improvement in colonic barrier function [150]	IL-10 Knock out mice
	Inhibition of TNF-α induced IL-8 secretion, MAPK, and NF-Κβ activation in HT-29 cells [151]	HT-29 cells
	Potentiation of mucin expression [152]	LSL174T cells and in vivo in rats
	No repair in colonic barrier breakdown or attenuation of colitis [153]	DSS-induced colitis in mice
	Effective against maintenance and treatment of active UC [154]	Clinical Trial
**Prebiotic Intervention**	**Effect on inflammation**	**Type of study/species**
GBF (Germinated barley foodstuff) [155]	Improvement in microflora composition [156] Increase in butyrate levels [157] Decrease in serum IL-8 concentration and in α1-acid glucoprotein concentration [158] Suppression of mucosal infiltration from inflammatory cells [159]	DSS-induced colitis in rats
Fructo and milk oligosaccharides [160]	FOS: Attenuation of trinitrobenzenesulfonic acid (TNBS) induced colitis Increased presence of lactic acid-producing bacteria Increased butyrate levels [161]	TNBS-induced colitis in rats
	Fructooligosaccharide: No improvement in disease activity [162]	DSS-colitis model in mice
	Goat milk oligosaccharides: Beneficial in the maintenance of body weight in DSS-mice Decreased myeloperoxidase activity Increased MUC3 expression Milder disease manifestation [163]	DSS-colitis model in mice
	Fructooligosaccharide: Increased IL-10 expression in dendritic cells as well as the increase of *Bifidobacteria* [164]	Human Crohn’s disease patients
Inulin and oligofructose [165]	Oligofructose: Increased Bifidobacteria and Lactobacilli activity [166]	Healthy Humans
	Oligofructose: Prevention of colitis development [167]	HLA-B27 transgenic mice
	Inulin (separately): Attenuated inflammation and caused an increase in lactic bacteria presence with a decrease in pH [168]	DSS-colitis model in mice
	Inulin and oligofructose in combination: increase in lactic bacteria and decreased pH [166]	Healthy Humans
Psyllium [169]	Amelioration of colonic damage through increased SCFA synthesis Decreased synthesis of pro-inflammatory cytokines [170]	-HLA-B27 transgenic rats
**Synbiotic Intervention**	**Effect on inflammation**	**Type of study/species**
Bifidobacterium longum and inulinoligofructose	Beneficial synergistic effect Improved disease pathology in comparison to prebiotics or probiotics by themselves Decrease in TNF-α and IL-1b levels Anti-inflammatory effect seen in endoscopic markers [171]	Clinical trial with UC patients
Bifidobacterium longum and psyllium	Beneficial synergistic effect Improved disease pathology in comparison to prebiotics or probiotics by themselves [172]	Clinical trial with UC patients
B.breve Yakult strain and galactooligosaccharides	Decrease in inflammation in mild to moderate UC [173]	Clinical trial with UC patients

## 6. Future Perspectives

As recapitulated in Figure 3, it is in the authors’ opinion that the use of biotics, with their demonstrated improvements in gut inflammation, may be a valuable therapeutic target for the treatment of CKD, especially with the need for alternatives to indoles-related strategies. Even though this has, as of yet, not been demonstrated in patients with CKD, well-designed studies, with a long duration and a variety of outcomes (e.g., microbiome composition, renal function, cardiovascular parameters, inflammatory markers), on a large population sample should be conducted. Furthermore, more research needs to be conducted into the precise mechanisms by which beneficial indole catabolites can be manipulated to reach eubiosis, improve intestinal homeostasis and reduce the progression of CKD. Indole metabolites also represent an attractive therapeutic target, although currently, they do represent a field where much still remains to be elucidated and consolidated. Contrastingly, SCFAs as a therapeutic option, have shown significant benefits in gastrointestinal inflammation management clinically.

## 7. Conclusions

Our knowledge of interactions of the microbiota with the host community has significantly increased over the past years, with the extent of the impact that tryptophan catabolites, especially, have on local and systemic inflammation becoming more apparent. However, further research is needed to fully consolidate the roles of some tryptophan catabolites as well as their mechanisms of action and how affecting these would alter inflammation and other CKD-associated comorbidities. Regardless, the manipulation of the microbiota to generate a more eubiotic composition through the use of “biotics” represents a very attractive therapeutic target, which is gaining a lot of traction, especially with the existing knowledge of how SCFAs positively affect both CKD and inflammation. The conduction of further well-designed randomised placebo-controlled clinical trials with long intervention times is an absolute necessity to enhance the understanding of how these biotics influence the maintenance of intestinal homeostasis as well as whether they can be therapeutically used for the treatment of CKD.

## Figures and Tables

**Figure 1 toxins-14-00645-f001:**
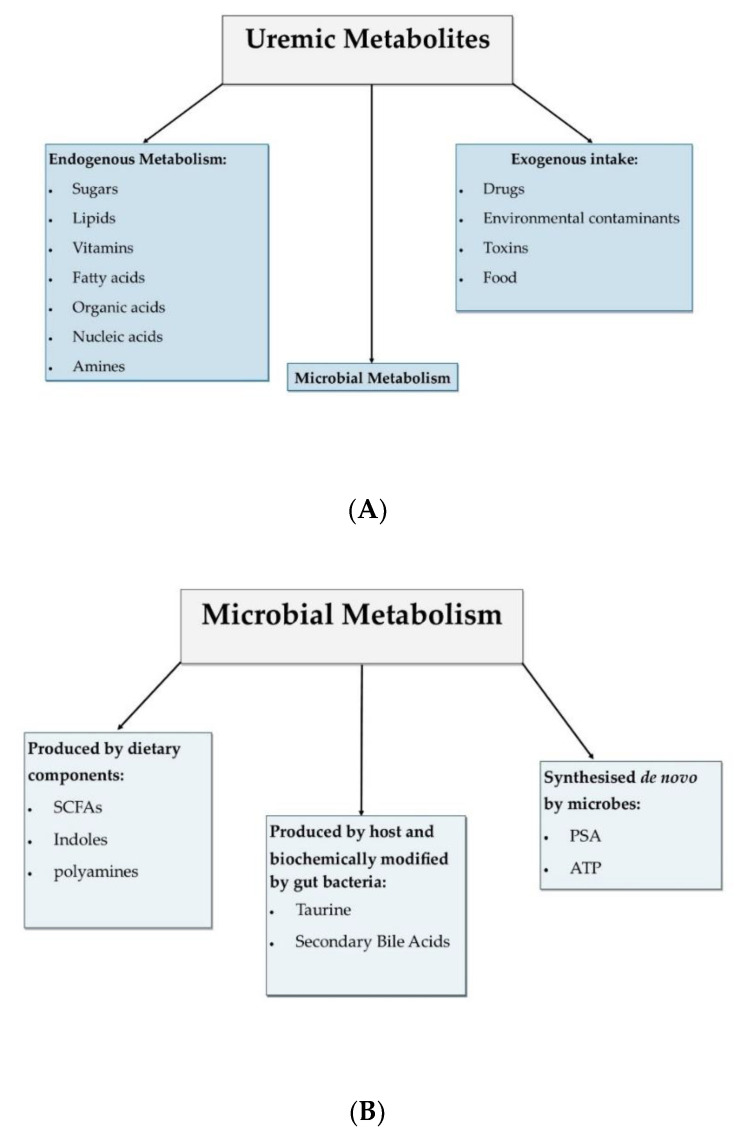
(**A**). Classification of uremic metabolites based on origin. Uremic metabolites have three possible origins, either endogenous metabolism, microbial metabolism, or from exogenous intake. (**B**). Classification of uremic metabolites sourced from microbial metabolism. Uremic metabolites sourced from microbial metabolism are further classified according to the involvement of microbiota in their synthesis; they are either the result of dietary component metabolism, produced by host and biochemically modified by microbiota or synthesised de novo by the microbiota. SCFAs: short chain fatty acids; PSA: polysaccharide-A [4].

**Figure 2 toxins-14-00645-f002:**
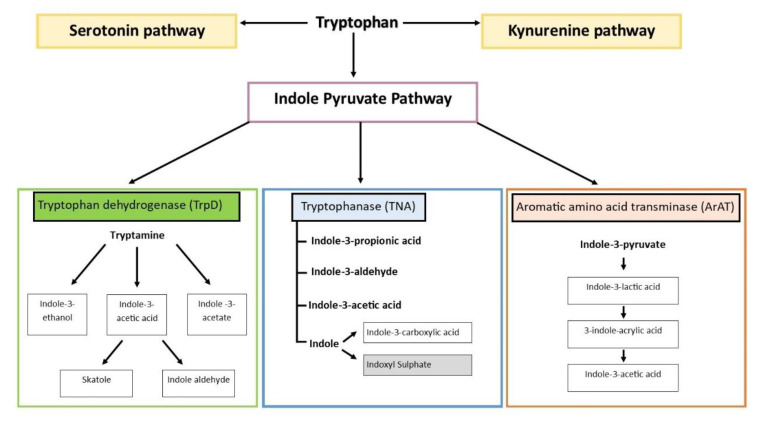
Depiction of metabolic pathways of tryptophan. Serotonin, and kynurenine pathways (in yellow boxes) are processes occurring endogenously within host metabolism. Indole pyruvate pathway occurs intestinally as a result of enzymatic degradations performed by the microbiota population in three separate pathways; tryptophan dehydrogenase pathway, tryptophanase pathway, and the aromatic amino acid transaminase pathway. All catabolites are produced intestinally with the exception of indoxyl sulfate (in the grey box), which is produced hepatically [34].

**Figure 3 toxins-14-00645-f003:**
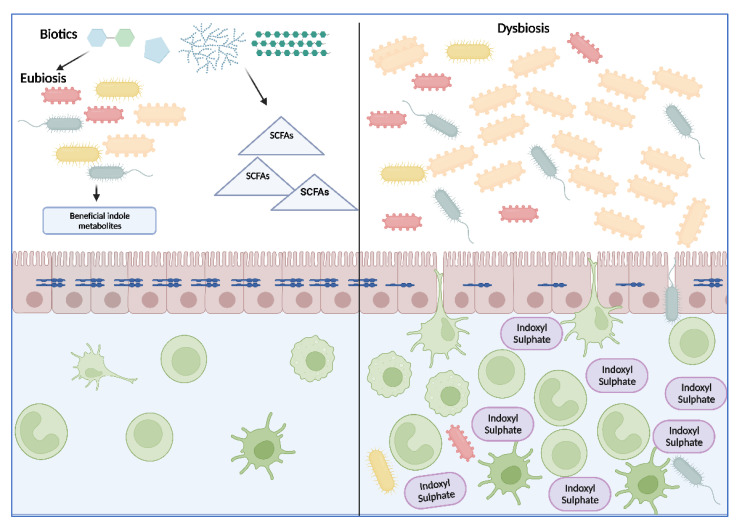
Depiction of future experimental perspectives. The use of biotics represents an important therapeutic target to achieve eubiosis and reduce intestinal inflammation, either through the promotion of the synthesis of beneficial indole metabolites or through the enhanced synthesis of SCFAs. A decrease in the levels of deleterious metabolites such as indoxyl sulfate is warranted for the reduction of intestinal homeostasis.

**Table 1 toxins-14-00645-t001:** Summary of AhR and PXR interaction of intestinally synthesised tryptophan catabolites in human AhR reporter assays. IE: indole-3-ethanol; IAA: indole-3-acetic acid; Skatole: 3-methylindole; IAld: indole-3-aldehyde; IPA: indole-3-propionic acid; IPγA: indole-3-pyruvate; ILA: indole-3-lactic acid; IAcr; indoleacrylic acid.

Catabolite	AhR interaction			PXR interaction		
	Affinity	Potency	Efficacy	Affinity	Potency	Efficacy
Tryptamine	Very Low	Low	Medium	Unknown	Very Low	Low
IAA	Very Low	Low	Very Low	Unknown	Very Low	Very Low
IE	Low	Low	Low	Unknown	Very Low	Low
Skatole	Low	Low	Very Low	Unknown	Very Low	Very Low
IAld	High	Low	Low	Low	Medium	High
IPA	Very Low	Low	Very Low	Unknown	Very Low	Very Low
Indole	Very Low	Low	Very Low	Unknown	Very Low	Very Low
IPγA	Very Low	Very Low	Very High	Low	Very Low	High
ILA	High	Low	Very High	Unknown	Very Low	Very Low
IAcr	Very Low	Very Low	Very Low	Unknown	Very Low	Very Low

Affinity: quantification of compound binding at the receptor–ligand binding domain; Potency: concentration of compound required to produce half-maximal effective effect; Efficacy: maximum response that can be generated as a result of ligand binding [35,36].

## Data Availability

Not applicable.

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
