# Peer review of "Microbial-Derived Tryptophan Catabolites, Kidney Disease and Gut Inflammation"

_toxins, 2022, doi:10.3390/toxins14090645_

Round 1
Reviewer 1 Report
The authors review uremic metabolites synthesized by bacteria from dietary sources with a focus on indole and its derivatives. The article is well written, detailed and clear despite the complexity of the subject. The topic is of interest to researchers in related fields. Nevertheless, some changes and additions should be made.
Please explain the abbreviations SCFAs and PSA also in the legend of Figure 1.
Lines 159 to 162: The harmful effects of IS should be mentioned in this paragraph “These indole products…” as not all of them are beneficial.
The following substances in Figure 2 are not discussed in the text (section 4) and table 2: indole-3-ethanol (explained in lines 346-347: “not have established roles in CKD or intestinal inflammation”), indole-acetate, indole-3-carboxylic acid, indoxyl sulfate (in table 2 but not in the text; after line 345). This discrepancies should be clarified/discussed.
Table 1: It should be stated whether the data refer to human or mouse AhR considering that “There exists a significant difference between affinity and activation between mouse and human AhR which could result in a 10-fold difference in ligand-binding affinity between the mouse AhR and human AhR for a diverse range of AhR ligands” (lines 225-227).
IAA (indole-3-acetic acid) is mentioned in the legend to table 1 but does not appear in the table.
Lines 229-230: “While all catabolites tested were capable of dose-dependently binding to and activating AhR and PXR, not all of them were able to induce the expression of … sensitive markers for activation…” Given that the activation markers were not induced, what were the criteria for activation?
I cannot follow the explanation of this discrepancy (lines 232 ff.).
Lines 237-238 (…specific catabolites also showed strong antagonistic properties against one or more of the prototypical AhR-ligands…). Which specific catabolites?
Lines 310-311: “IL-22 secretion IAld inhibited TNF-α and IL-1β secretion, which..” This sentence does not appear to be complete.
Even though “the definition of postbiotics is a matter of dispute” (line 407), a brief comment on possible definitions would be interesting in this context.
If it is compatible with the guidelines of the journal, a clearer presentation of the tables would be desirable.
Author Response
Dear Reviewer, thank you very much for taking the time to read through my manuscript and your kind comments and suggestions on how to improve it. Please find a point-by-point response:
- Reviewer’s point: Please explain the abbreviations SCFAs and PSA also in the legend of Figure 1.
- Author’s response: Please note these have been now included in the legend of figure 1, as denoted in red.
- Reviewer’s point: Lines 159 to 162: The harmful effects of IS should be mentioned in this paragraph “These indole products…” as not all of them are beneficial.
- Author’s response: The text has been altered to read ‘’These indole products, with the exception of IS, are of importance for the microbial community in drug resistance, biofilm formation, toxicity, plasmid stability and spore formation [29-32]’’ to denote that not all indole derivatives do possess a beneficial role.
- Reviewer’s point: The following substances in Figure 2 are not discussed in the text (section 4) and table 2: indole-3-ethanol (explained in lines 346-347: “not have established roles in CKD or intestinal inflammation”), indole-acetate, indole-3-carboxylic acid, indoxyl sulfate (in table 2 but not in the text; after line 345). This discrepancies should be clarified/discussed/
- Author’s response: Section 4 (line 353) has been altered to include ‘’According to our knowledge, the tryptophan catabolites indole-3-acetamide and indole-3-ethanol, indole acetate and indole-3-carboxylic acid do not have established roles in CKD or intestinal inflammation.’’ To our knowledge, findings to suggest roles for these metabolites in CKD or intestinal inflammation has not been published. Additionally, the effects of indoxyl sulfate have been discussed in line 345 hence in the author’s belief, discussing these effects again would be an unnecessary repetition.
- Reviewer’s point: Table 1: It should be stated whether the data refer to human or mouse AhR considering that “There exists a significant difference between affinity and activation between mouse and human AhR which could result in a 10-fold difference in ligand-binding affinity between the mouse AhR and human AhR for a diverse range of AhR ligands” (lines 225-227).
- Author’s response: The table has been amended in light of this point.
- Reviewer’s point: IAA (indole-3-acetic acid) is mentioned in the legend to table 1 but does not appear in the table.
- Author’s response: We sincerely apologise for this oversight and have amended the table to also include IAA.
- Reviewer’s point: Lines 229-230: “While all catabolites tested were capable of dose-dependently binding to and activating AhR and PXR, not all of them were able to induce the expression of …sensitive markers for activation…” Given that the activation markers were not induced, what were the criteria for activation? cannot follow the explanation of this discrepancy (lines 232 ff.)
- Author’s Response: We’ve adapted the text to read ‘’While all catabolites tested were capable of dose-dependently binding to and activating AhR and PXR in luminescence-based reporter assays, not all of them were able to induce the expression of CYP1A1 mRNA, a sensitive marker for AhR-activation, or CYP3A4 mRNA, a sensitive marker for PXR-activation. This discrepancy could potentially be explained by the interactions between AhR and PXR. Activation of PXR has been shown to prevent binding of AhR to its target genes and subsequently suppressing the gene expression [57].’’ Essentially what happens is that AhR may become activated as shown in luminescence-based reporter assays, but it may be prevented from binding on its target genes as a result of PXR binding thereby preventing AhR from binding. We hope that this clarifies the discrepancy.
- Reviewer’s point: Lines 237-238 (…specific catabolites also showed strong antagonistic properties against one or more of the prototypical AhR-ligands…). Which specific catabolites?
- Author’s response: We have adapted the text to read ‘’Antagonism was observed at the level of AhR activation but also at the level of inhibition CYP1A1 expression. The catabolite indole proved to be the most proficient antagonist of AhR-activation confirming previously obtained results using a different methodology [36, 58]. Antagonistic activity has been shown by other tryptophan catabolites, but results are rather ambiguous at this stage.‘’ We hope that this is sufficient.
- Reviewer’s point: Lines 310-311: “IL-22 secretion IAld inhibited TNF-αand IL-1β secretion, which..” This sentence does not appear to be complete.
- Author’s response: We have deleted this sentence on its entirety as it was unclear and not of added value.
- Reviewer’s point: Even though “the definition of postbiotics is a matter of dispute” (line 407), a brief comment on possible definitions would be interesting in this context.
- Author’s response: We have altered the text to include the definition ‘’Postbiotics also represent a biotic intervention option and have recently been defined by the international scientific association of prebiotics and probiotics as a preparation of inanimate microorganisms and/or their components that confers a health benefit on the host. Assessments on the efficacy of postbiotics on intestinal inflammation or the management of CKD have been very limited and hence postbiotics will not be discussed in this review. ‘’
- Reviewer’s point: If it is compatible with the guidelines of the journal, a clearer presentation of the tables would be desirable.
- Author’s response: Thank you very much for this comment, we agree, but unfortunately we have to follow the journal guidelines on the matter and cannot make any of the tables clearer without completely changing the formatting style.
We hope we have answered all your points sufficiently and would like to thank you for all your time to go through this manuscript and for providing us with this feedback.
Reviewer 2 Report
This review is complete and detailed. Only some formats (Table 3 and reference 87) and styles need revision.
Author Response
Dear Reviewer, thank you very much for taking the time to read through our manuscript and your kind comments and suggestions on how to improve it. Please be aware we have adapted the formatting change for the reference which you highlighted but we are unable to adapt the tables formatting as this would interfere with the guidelines of the journal.
We hope we have answered your point sufficiently and would like to thank you once again for all your time to go through this manuscript and of course the feedback.
Reviewer 3 Report
The authors submit an interesting review article entitled "Microbial-derived tryptophan catabolites, kidney disease and gut inflammation".
They discuss in details the roles of tryptophan catabolites in gut inflammation and chronic kidney disease.Interestingly they also devote a strong part of their review to describe the use of biotics to achieve intestinal homeostasis and alleviate toxicity due to uremic toxins.
The review is well written and comprehensive. Perhaps when the authors discuss the role of AhR, they can describe what happens the various knock-out murine models, concerning uremic toxin accumulation.
Also at least one recapitulative figure would be useful to the reader.
Author Response
Dear Reviewer, thank you very much for taking the time to read through our manuscript and your kind comments and suggestions on how to improve it. Please find a point-by-point response:
- Reviewer’s point: Perhaps when the authors discuss the role of AhR, they can describe what happens the various knock-out murine models, concerning uremic toxin accumulation.
- Author’s response: We have added evidence for the potential use of AhR antagonism as a therapeutic target evidenced by murine AhR KO models in lines 397-400 (in red). We hope this is sufficient.
- Reviewer’s point: Also at least one recapitulative figure would be useful to the reader.
- Author’s response: We have added a figure which we believe recapitulates our entire review at the end of our manuscript.
We hope we have answered your point sufficiently and would like to thank you once again for all your time to go through this manuscript and of course the feedback.